# REPO: Language Models with Context Re-Positioning

**Huayang Li** [1 2]   **Tianyu Zhao** [1]   **Deng Cai** [3]   **Richard Sproat** [1]

## Abstract

In-context learning is fundamental to modern Large Language Models (LLMs); however, prevailing architectures impose a rigid and fixed contextual structure by assigning linear or constant positional indices. The rigid position information poses the full burden of organizing the input structure to attention layers, thus reducing the amount of attention that could be allocated for more critical information. To address this, we propose REPO, a novel mechanism that alleviates the burden for attention layers via context re-positioning. Unlike conventional approaches, REPO utilizes a differentiable module, $f_\phi$, to assign token positions that capture contextual dependencies, rather than replying on pre-defined order. By continually pre-training on the OLMo-2 1B & 7B models, we demonstrate that REPO consistently enhances performance on tasks involving noisy contexts, structured data, and longer context length, while maintaining competitive performance on general short-context tasks. Analysis reveals that REPO successfully allocates more attention mass to distant but relevant information, assigns positions in a dense and non-linear space, and captures the intrinsic structure of the input context. Our code is at `https://github.com/SakanaAI/repo`.

## 1. Introduction

The emergence of large language models (LLMs) (Brown et al., 2020) has enabled a wide range of applications, including few-shot learning (Brown et al., 2020), retrieval-augmented generation (Lewis et al., 2020; Yao et al., 2023), and agentic systems (Schick et al., 2023; Park et al., 2023). At the core of these applications is in-context learning (Shen et al., 2024), a form analogous to human *working memory*,

---

[1]Sakana AI, Japan [2]Nara Institute of Science and Technology (NAIST), Japan [3]Independent Researcher. Correspondence to: Huayang Li <li.huayang.lh6@is.naist.jp>, Tianyu Zhao <tianyu@sakana.ai>, Richard Sproat <rws@sakana.ai>.

*Proceedings of the $43^{rd}$ International Conference on Machine Learning*, Seoul, South Korea. PMLR 306, 2026. Copyright 2026 by the author(s).

where information within a limited context window is temporarily stored and processed to solve a task. Consequently, exploring how to effectively utilize context information has become a fundamental research line in the LLM era (Wei et al., 2022; Weston & Sukhbaatar, 2023; Chen et al., 2023).

Recent studies show that an LLM's ability to leverage contextual information is strongly influenced by its position encoding scheme (Vaswani et al., 2017; Press et al., 2021; Su et al., 2024). Most LLMs impose a fixed contextual structure by assigning tokens consecutive integer indices from 0 to $L-1$ (Vaswani et al., 2017) or a constant index $a$ for all tokens (Kazemnejad et al., 2023). These indices are then integrated into a model through position encoding functions, enforcing a rigid organization of context.

Although fixed position assignments have become the *de facto* standard, they deviate from how human working memory processes information. Studies on human learning show that the presence of structural information facilitates text processing (Hyönä & Lorch, 2004; Schneider et al., 2018). However, the capability of turning linearized text into a structured representation is absent from the architectural design of modern LLMs. As a result, the attention mechanism is fully responsible for understanding the underlying language structure of linearized texts. This burden impairs the computational allocation for contextual reasoning, thus harming model performance in tasks that require strong long-range or fine-grained contextual dependencies, e.g., needle-in-a-haystack (NIAH) problems (Kamradt, 2023) or question answering under highly diluted contexts (Hsieh et al.). This can be additionally interpreted in terms of Cognitive Load Theory (CLT) (Sweller, 1994; Paas et al., 2003), which states that the capacity of working memory is influenced by the extraneous load (how information is organized and presented) and germane load (the learning process). While the task difficulty and model capacity is fixed, an attention layer has to assign its limited capacity to understanding the structure of an input sequence and processing the input. The performance of the latter degrades when the former consumes too much capacity.

We propose an internal mechanism for LLMs to reduce *extraneous load* by re-organizing the positions of tokens. We formalize this process, termed *context Re-Positioning* (REPO), as learning non-constrained position values based

on information relevance of tokens, in contrast to using fixed linear positions in prior work. To this end, we introduce a differentiable module $f_\phi$, which assigns a position value in continuous space for each token based on its hidden state. The $f_\phi$ can be independently learned for each attention head of an LLM. Trained on general data, $f_\phi$ learns to re-position tokens free from conventional constraints like monotonicity or integer values. The continuity of modern position encoding functions, e.g., RoPE (Su et al., 2024) and ALiBi (Press et al., 2021) methods, is key to the end-to-end optimization of $f_\phi$, as it allows these assigned positions to be integrated in a differentiable manner.

We find that LLMs trained with the REPO method achieve consistent performance gains on tasks involving noisy context, structured data, and long contexts. In our experiments, we continually pre-train LLMs using REPO and several baseline approaches based on the fully open-sourced OLMo-2 1B and 7B models, in order to avoid biased or misleading results arising from data contamination issues (Wu et al., 2025; Dong et al., 2024). On OLMo-2 1B, REPO outperforms RoPE by at least 5.4, 2.2, and 6.9 points on benchmarks targeting noisy context, structured data, and long context, respectively (Bai et al., 2024). We then evaluate REPO on OLMo-2 7B to assess its effectiveness at larger model scales. The results show similarly consistent performance gains, indicating that REPO scales well with increased model capacity. In addition to these improvements, REPO achieves comparable performance on a wide range of general benchmarks (Wang et al., 2024b; Clark et al., 2018; Zellers et al., 2019), which are typically short-context and require minimal reorganization.

To better understand whether our method aligns with the CLT-based interpretation and where the performance gains of the REPO method come from, we conducted a series of detailed analyses. **First**, to evaluate how different methods handle long-range dependencies, we compared the attention distributions across methods, particularly focusing on their treatment of distant but relevant information. In the NIAH task (Section 5.1), we observed that REPO allocates more attention to the most critical "needle" tokens compared to the baseline methods, while directing less attention to the nearest "query" tokens. This behavior breaks the typical locality bias (Yang et al., 2025b; Press et al., 2021), dynamically adjusting based on the context. **Second**, the positions assigned by REPO exist in a denser, more non-linear space, which is critical for enhancing its generalization to longer contexts, such as extending from 4K to 16K tokens. Finally, an interesting finding is that REPO learns positional patterns that resemble a hybrid of prior position-assignment strategies. These include assigning constant positions $a \pm \epsilon$, as in NoPE (Kazemnejad et al., 2023), and enforcing a monotonic position sequence, as in RoPE (Su et al., 2024) and other position encoding methods (Vaswani et al., 2017; Press et al.,

2021), within a given context span (Section 5.2). In our case study (Appendix D), we also found that the positions assigned by REPO capture the intrinsic structure of the input context (e.g., segmentation of few-shot examples).

**Conflict of Interest Disclosure**  The authors declare that they have no known competing financial interests or personal relationships that could have appeared to influence the work reported in this paper.

## 2. Background

The position information of tokens in a context is critical for the attention mechanism of LLMs. The position of a given token is generally mapped into embeddings (Su et al., 2024; Vaswani et al., 2017; Li et al., 2025) or biases (Press et al., 2021; Raffel et al., 2020) through a position encoding module before the attention function. In this section, we will introduce the notations for attention mechanism and position encoding functions.

Given an input sequence $\boldsymbol{x} = (x_1, x_2, \ldots, x_L)$, where each token $x_i$ is drawn from a vocabulary $\mathcal{V}$, most large language models (LLMs) process the information in $\boldsymbol{x}$ through multiple layers of self-attention-based neural networks. In each layer, the attention score[1] between tokens $x_i$ and $x_j$ is computed as follows:

$$\mathbf{Q}, \mathbf{K}, \mathbf{V} = \mathbf{H}\mathbf{W}^{q,k,v}, \tag{1}$$

$$\mathbf{A}_{i,j} = \boldsymbol{q}_i^\top \boldsymbol{k}_j, \tag{2}$$

where $\mathbf{W}^{q,k,v} \in \mathbb{R}^{d \times 3d}$ projects the hidden state $\boldsymbol{h}_i \in \mathbf{H}$ of token $x_i$ into its corresponding query, key, and value representations, and $\mathbf{A} \in \mathbb{R}^{L \times L}$ represents the attention scores for all token pairs.

We use the *rotary position encoding* (RoPE) (Su et al., 2024) as a specific example to illustrate the position assignment and encoding procedure commonly employed in LLMs (Walsh et al.; Yang et al., 2025a; Qwen et al., 2025). First, each token $x_i$ is assigned an integer position index $i$. This positional information is then incorporated into the model through an encoding function $g_\theta : \mathbb{R} \to \mathbb{R}^{d \times d}$, which is a differentiable function that generates a rotation matrix. The parameter $\theta$ represents a pre-defined frequency vector, generally frozen during training. In RoPE, the function $g_\theta$ is directly applied to the query $\boldsymbol{q}_i$ and key $\boldsymbol{k}_j$ as follows:

$$\mathbf{A}_{i,j}^{\text{RoPE}} = \boldsymbol{q}_i^\top g_\theta(j - i) \boldsymbol{k}_j, \tag{3}$$

where $i$ and $j$ are the integer position indices for tokens $x_i$ and $x_j$, respectively, and $g_\theta(j - i)$ captures the relative

---

[1]Here, we only discuss the single-head scenario for simplicity. However, the attention score can be easily extended to the multi-head variant.

positional relationship between any tokens with a distance of $j - i$.

In many related studies, strict linear position assignment (i.e., 0 to $L - 1$) similar to RoPE has become the standard approach, with various encoding functions being explored (Vaswani et al., 2017; Press et al., 2021; Yang et al., 2025b; Li et al., 2025). One notable exception is the NoPE method (Kazemnejad et al., 2023; Yang et al., 2025b), which omits both position assignment and position encoding. However, we demonstrate that this approach is equivalent to applying position encoding at a constant position $a$ for all tokens. Further discussion about the inter-connection between constant (e.g., NoPE) and linear position assignment (e.g., RoPE) strategies are provided in Appendix A.

## 3. Methods

### 3.1. Overview

The current linear context organization, which assigns consecutive integer indices to tokens, overlooks the internal structure of tokens based on their relevance. For instance, this limitation leads to noticeable performance degradation in tasks involving long-distance dependencies (Hsieh et al.), a problem analogous to the high extraneous load issue in Cognitive Load Theory (CLT).

The main goal of this work is to reduce the unnecessary cost raised by the oversimplified organization of the context, i.e., *extraneous cognitive load*, thereby conserving the finite working memory capacity for beneficial *germane load*, such as the attention mechanism. To this end, we propose a context re-positioning (REPO) module $f_\phi : \mathbb{R}^d \to \mathbb{R}$, which is a light-weight neural network that assigns more appropriate positions of tokens, taking into account their relevance within a given context. The assigned positions by $f_\phi$ are defined in a continuous, non-linear space, and can thus be optimized jointly with LLMs when equipped with differentiable position encoding methods (Vaswani et al., 2017; Su et al., 2024; Press et al., 2021). Notably, our REPO module $f_\phi$ is prior to position encoding, where the latter aims to map assigned positions into embeddings or biases. An illustration for REPO is shown in Figure 1.

### 3.2. Context Re-positioning

The context re-positioning module $f_\phi$ has two components: 1) representing the position information based on hidden states of tokens; 2) assigning real-valued positions based on the extracted position representations.

**Position Representation**  Kazemnejad et al. (2023) shows that position information may be entangled in original hidden states, so the first component is designed to extract the position representation from hidden states of tokens explic-

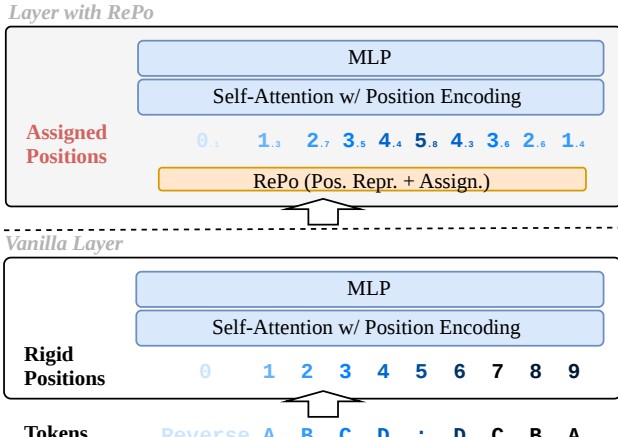

*Figure 1.* Illustration of the differences between REPO and the vanilla method using the reversal task (Appendix C). The positional encoding can be any standard method (e.g., RoPE, ALiBi). All layers take the hidden states of the token sequence as input.

itly. In our implementation, we use a light-weight SwiGLU (Shazeer, 2020) sub-layer to achieve this goal:

$$\boldsymbol{r}_i = \mathrm{Swish}(\boldsymbol{h}_i \mathbf{W}^g) \odot (\boldsymbol{h}_i \mathbf{W}^c), \tag{4}$$

where $\boldsymbol{r}_i \in \mathbb{R}^{d_p}$ and $\boldsymbol{h}_i \in \mathbb{R}^d$ are position representation[2] and hidden state of token $x_i$ respectively, $\mathbf{W}^g, \mathbf{W}^c \in \mathbb{R}^{d \times d_p}$ are linear transformations for gate and content mapping, and $\mathrm{Swish}(\cdot)$ is the activation function. Since we assume that position information can be represented with a lower dimension, we set $d_p < d$ in practice.

**Position Assignment**  The subsequent component assigns a new position value $z_i$ for token $x_i$ in each attention head. There are a variety of modeling strategies for processing $\boldsymbol{r}_i$ with full (Vaswani et al., 2017) or limited (LeCun & Bengio, 1998) access to $\boldsymbol{r}_{<i}$, but we find that a linear transformation achieves comparable performance with lower latency:

$$z_i = \boldsymbol{r}_i \mathbf{W}^z, \tag{5}$$

where $\mathbf{W}^z \in \mathbb{R}^{d_p \times 1}$.

**REPO Module**  By combining Eq. 4 and 5, the formal definition of $f_\theta$ becomes:

$$f_\phi(\boldsymbol{h}_i) = \big(\mathrm{Swish}(\boldsymbol{h}_i \mathbf{W}^g) \odot (\boldsymbol{h}_i \mathbf{W}^c)\big) \mathbf{W}^z, \tag{6}$$

where $\boldsymbol{h}_i$ is the hidden state of token $x_i$, as defined in Eq. 1. When equipped with modern position encoding methods,

---

[2]Notably, the hidden state $\boldsymbol{h}_i$ of token $x_i$ does not explicitly encode the linear position information $i$. In our preliminary experiments, when using the raw position $i$ as an additional dimension during continue-training, the LLM pre-trained on RoPE quickly biases to this feature, resulting in trivial position assignment.

*Table 1.* Performance on noisy context. We evaluate on needle-in-a-Haystack (NIAH) tasks in RULER benchmark (Hsieh et al.) with four types of needles: single (S), multi-query (MQ), multi-key (MK), and multi-value (MV). The best results are in **bold** text.

| Base | Method | S | MV | MK | MQ | AVG. | Δ |
|------|--------|-----|------|------|------|------|------|
| | RoPE | **100.0** | 78.5 | **86.0** | 79.0 | 85.9 | 0.0 |
| | NoPE | 83.0 | 83.3 | 76.0 | 78.5 | 80.2 | −5.7 |
| 1B | R2N1 | **100.0** | **91.8** | 82.0 | 88.3 | 90.5 | +4.6 |
| | N2R1 | 95.0 | 89.8 | 84.0 | 84.3 | 88.3 | +2.4 |
| | REPO | **100.0** | 89.3 | **86.0** | 89.8 | **91.3** | **+5.4** |
| 7B | RoPE | **100.0** | 97.3 | **97.0** | 94.8 | 97.3 | 0.0 |
| | REPO | **100.0** | **100.0** | 96.0 | **95.5** | **97.9** | **+0.6** |

*Table 2.* Performance for structured data. We evaluate on HybridQA dataset (Chen et al., 2020) for table data. We use exact match (EM) as metric. We highlight the best results with **bold** text.

| Base | Method | HQA-Table | Δ vs RoPE |
|------|--------|-----------|-----------|
| | RoPE | 24.43 | 0.00 |
| | NoPE | 23.52 | −0.91 |
| 1B | R2N1 | 25.11 | +0.68 |
| | N2R1 | 23.86 | −0.57 |
| | REPO | **26.70** | **+2.27** |
| 7B | RoPE | 33.52 | 0.00 |
| | REPO | **37.61** | **+4.09** |

e.g., RoPE, the computation of attention score becomes:

$$\mathbf{A}_{i,j}^{\text{RePo}} = \boldsymbol{q}_i^\top g_\theta\big(f_\phi(\boldsymbol{h}_j) - f_\phi(\boldsymbol{h}_i)\big)\boldsymbol{k}_j$$
$$= \boldsymbol{q}_i^\top g_\theta(z_j - z_i)\boldsymbol{k}_j, \tag{7}$$

where the position encoding function $g_\theta$ is the same as that used in RoPE in Eq. 3. It is worth noting that our REPO is not restricted to RoPE and can be easily extended to all the differentiable position encoding methods (Vaswani et al., 2017; Press et al., 2021; Li et al., 2025). In practice, we apply the position representation module (Eq. 4) for each layer independently, and position assignment module (Eq. 5) for each attention head independently.

**Training & Efficiency** Since many position encoding methods are differentiable (Su et al., 2024; Press et al., 2021; Li et al., 2025), we can train an LLM with REPO-based attention (Eq. 7) using backpropagation when equipped with such encodings. To balance efficiency and effectiveness, we apply the REPO method starting from the $\text{ceil}(L/3)$-th layer of LLMs while keeping standard position encoding for the lower layers, where $L$ is the number of layers. This design choice is motivated by previous findings that the lower layers of LLMs primarily capture surface-level features that depend more on local information (Tenney et al., 2019), such as part-of-speech tagging and syntax, and thus benefit less from reorganization. An ablation study for the hyper-parameter is in Appendix B.5.

In order not to impair the efficiency of LLMs significantly, we only use the assigned position $z_i$ and $z_j$ to affect the position encoding in attention calculation in Eq. 7, leaving the auto-regressive order of $\boldsymbol{q}_i$ (or $\boldsymbol{k}_i$) and $\boldsymbol{q}_j$ (or $\boldsymbol{k}_j$) in the context unchanged. The REPO module can be applied to each attention head independently. In principle, we can sort queries and keys in each attention head according to the assigned positions $\boldsymbol{z} = \{z_1, \ldots, z_n\}$. However, under the auto-regressive language modeling paradigm, this approach requires re-computation for the KV cache at each time step, resulting in tremendous overhead for auto-regressive LLMs. Therefore, the assigned positions are only used in Eq. 7.

## 4. Experiments

This section presents our main experiments on general language modeling. We continually pre-train an LLM with REPO on general datasets and evaluate its performance on three types of tasks that require restructuring context. In Appendix C, we present preliminary studies on synthetic data along with visualizations to show how REPO works.

### 4.1. Setup

We use OLMo-2 (Walsh et al.) as the backbone model, which is fully open-sourced, including its training data, model weights, and code. Our primary motivation for conducting experiments on a fully open-source LLM is to avoid data contamination issues (Wu et al., 2025; Dong et al., 2024), which can otherwise lead to biased or misleading results. Notably, OLMo-2 is trained with the RoPE method with linear position assignment (Su et al., 2024). For all methods in our experiments, we start from the checkpoint of the OLMo-2 1B and 7B model that have completed stage-1 pre-training on 4 trillion tokens, and we continually pre-train it on the 50B-token stage-2 data[3]. The training context length is 4096 tokens. We keep the training configuration and codebase identical to those released by Walsh et al..

For our REPO method, we use the RoPE function (Su et al., 2024) to encode the dynamic positions assigned by $f_\phi$, as shown in Eq. 7. We apply it starting from the $\text{ceil}(L/3)$-th layer, where $L$ is the number of layers, of the OLMo-2 model[4], i.e., the 5th and 10th layers for the 1B and 7B models, respectively. In each layer that uses REPO, we share the parameters for the position representation transformation in Eq. 4, while learning the position assignment for each

---

[3] https://github.com/allenai/OLMo

[4] In our preliminary study, we tried different strategies to apply REPO. First, we apply REPO to very few layers, and the performance of RePo after training becomes very close to RoPE — that is, the context repositioning module doesn't impact the system significantly. Second, we tried the reverse strategy, i.e., applying REPO to the former $1/3$ layers and standard RoPE to the latter $2/3$ layers: The training becomes unstable under this setting.

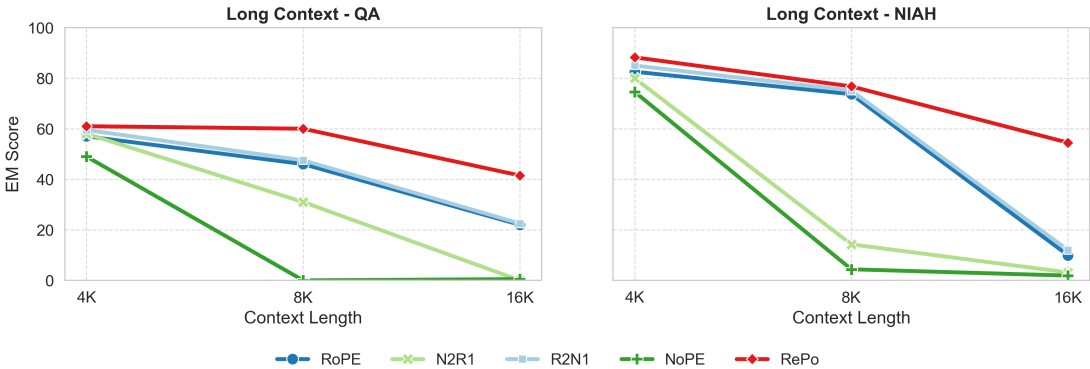

*Figure 2.* Long-context evaluation on OLMo-2 1B model. We use the data in RULER (Hsieh et al.) benchmark and apply YaRN (Peng et al., 2024) for all RoPE layers to extend the context. Similar results on OLMo-7B are shown in Appendix B.4.

attention head independently, as described in Eq. 5. The hidden size of the learned position representation is set to $1/8$ of the model's hidden size, based on the assumption that positional information is less rich than the information in the original hidden states. As shown in §5.4, REPO is lightweight and introduces negligible overhead compared to the original LLM.

We compare REPO with followings on OLMo-2 1B:

- ROPE: Uses RoPE (Su et al., 2024) for positional encoding, identical to pre-training the process (Walsh et al.). Unless otherwise specified, this method uses linear position assignment during training, i.e., from 0 to 4095.
- NOPE: Removes explicit positional encoding methods (Kazemnejad et al., 2023; Wang et al., 2024a), i.e., RoPE is omitted during continual pre-training. This is equivalent to constant position assignment (Appendix A).
- R2N1: A hybrid positional encoding method that interleaves RoPE and NoPE (Yang et al., 2025b). For every three layers, the first two use RoPE, while the last one uses NoPE.
- N2R1: The opposite baseline of R2N1.

We further evaluate REPO on the OLMo-2 7B model using ROPE as a baseline to demonstrate the effects of scaling up the model size. We train those models on 4 H100 GPUs for 50B tokens. We use the `allenai/olmes`[5] codebase for evaluation, which provides extensive test suites. We categorize our main evaluation tasks into three dimensions:

- **Noisy Context** evaluates the model's robustness to contexts containing large amounts of irrelevant information. Such "noise" increases *extraneous cognitive load* (Paas et al., 2003), which can negatively affect problem-solving performance (Peysakhovich & Lerer, 2023). We use multiple variants of needle-in-a-hystack (NIAH) tasks provided

by RULER benchmark (Hsieh et al.) for evaluation, which purposefully constructs contexts with irrelevant content for evaluation.

- **Structured Data** evaluates performance on structured data, e.g., tables. This setting highlights the importance of contextual organization, as linearizing such data into natural language often leads to significant structural information loss. We use HybridQA (Chen et al., 2020) to evaluate the performance on table reasoning.
- **Longer Context** evaluates model performance on sequences longer than those seen during training (i.e., 4K tokens), as positional encoding has been shown to strongly affect long-context generalization (Press et al., 2021). We use subsets of RULER and LongBench (Bai et al., 2024), which contain examples with 4K to 16K tokens.

Notably, both the evaluations for noisy context and structured data are conducted within the training context length (i.e., 4K tokens). In contrast, the longer-context evaluation requires extended context lengths. To enable this, we apply the YaRN method to RoPE layers across all methods using the extrapolation hyperparameters in Peng et al. (2024). More details are in Appendix B.

### 4.2. Results

Experimental results show that REPO yields significant performance gains across all three evaluation dimensions.

**Noisy Context**  All evaluation subtasks inject substantial irrelevant information into the input context. As shown in Table 1, REPO consistently improves performance under noisy conditions. On OLMo-2 1B, REPO achieves the best average accuracy (91.3), outperforming ROPE by 5.4 points and surpassing alternative re-positioning variants such as R2N1 and N2R1. When scaling to the 7B model, overall performance is already high across all methods; neverthe-

---

[5] https://github.com/allenai/olmes

*Table 3.* Performance on LongBench (2Wiki: *2WikiMultihopQA*, MSQ: *MuSiQue*, MFQA: *MultiFieldQA-en*, NQA: *NarrativeQA*, GovR: *GovReport*, TQA: *TriviaQA*). We report F1 score for QA and fewshot tasks and Rouge-L (Lin, 2004) for summarization tasks. We only evaluate on data that contains less than 16K tokens. Since the model is trained with a max of 4096 tokens, the YaRN method (Peng et al., 2024) is used for all RoPE layers for context extrapolation. We use **bold** for best results.

| Base | Method | Multidoc QA | | Singledoc QA | | Summarization | | Fewshot | AVG. | Δ vs RoPE |
| | | 2Wiki | MSQ | MFQA | NQA | GovR | QMSum | TQA | | |
|---|---|---|---|---|---|---|---|---|---|---|
| 1B | RoPE | 23.32 | 7.24 | 27.37 | 12.94 | 6.23 | 7.96 | 61.47 | 20.93 | 0.00 |
| | NoPE | 9.11 | 0.33 | 13.64 | 1.80 | 5.15 | 0.68 | 18.12 | 6.98 | −13.95 |
| | R2N1 | 25.88 | 7.31 | 31.28 | **16.24** | 4.53 | 8.74 | 66.67 | 22.95 | +2.02 |
| | N2R1 | 16.24 | 0.40 | 21.88 | 1.26 | 8.74 | 5.31 | 25.18 | 11.29 | −9.64 |
| | REPO | **30.86** | **13.45** | **33.12** | 15.24 | **16.80** | **12.53** | **73.02** | **27.86** | +**6.93** |
| 7B | RoPE | 22.10 | 6.69 | 32.61 | 16.38 | 11.19 | 6.63 | 83.97 | 25.65 | 0.00 |
| | REPO | **33.41** | **19.55** | **40.37** | **22.64** | **12.30** | **10.41** | **85.50** | **32.03** | +**6.38** |

less, REPO still yields consistent improvements, increasing the average score by 0.6 points over RoPE, Thus, improving 0.6 EM scores is non-trivial, i.e., 22% reduction of the remaining errors, and achieving perfect accuracy on the multi-value task. These results indicate that the re-positioning mechanism in REPO enhances robustness to noisy context, even when sequence lengths remain within the training regime.

**Structured Data**   Since linearizing structured data (e.g., tables) into natural language can result in substantial loss of structural information, it is of interest to examine whether contextual re-positioning benefits special data types. As shown in Table 2, REPO substantially improves exact-match accuracy on HybridQA compared to vanilla RoPE. On the 1B model, REPO outperforms RoPE by 2.27 points and exceeding all alternative baselines. The gains are even larger at the 7B scale, where REPO improves over RoPE by 4.09 points. The observations suggest that REPO better preserves and exploits latent structural cues in linearized table representations.

**Longer Context**   The advantage of REPO becomes even more pronounced on long-context tasks. As shown in Figure 2, REPO based on OLMo-2 1B already surpasses all baselines at a context length of 4K. The performance gap further widens at 8K and 16K, which are lengths unseen during training. To rule out potential confounding effects from REPO's noise robustness, we additionally evaluate on LongBench (Bai et al., 2024), which consists of more realistic long-context tasks. As reported in Table 3, REPO consistently outperforms RoPE across all task categories. On OLMo-2 1B, REPO improves the average LongBench score by 6.93 points over RoPE. Similar improvements are observed at the 7B scale, where REPO yields a 6.38-point average gain, demonstrating that the proposed re-positioning mechanism scales effectively with model capacity and supports robust generalization to extended contexts.

*Table 4.* Attention mass per token on different parts of context. We evaluate on NIAH task within the training context length (i.e., 4K tokens). Values are at the scale of $10^{-2}$.

| Pos. Assignment | Needle | Query | Rest |
|---|---|---|---|
| Linear (e.g., RoPE) | 1.754 | 1.123 | 0.014 |
| Constant (e.g., NoPE) | 1.572 | 1.135 | 0.014 |
| REPO | 2.013 | 1.046 | 0.015 |

## 5. Analyses

This section aims to provide insights into the inner workings of REPO. To this end, we conducted detailed analyses on the 1B model to understand: 1) where the performance gains stem from; and 2) what patterns the positions assigned by REPO exhibit.

### 5.1. Attention Mass on Relevant Tokens

Since our REPO method re-organizes the context based on its intrinsic structure, we hypothesize that it can better capture long-distance dependencies by bringing distant but relevant tokens with closer positions. To evaluate this effect, we analyze the attention patterns of methods with different types of position assignment strategies on the needle-in-a-haystack (NIAH) task (Kamradt, 2023; Hsieh et al.) and quantitatively measure the attention mass, i.e., attention scores averaged across attention heads and layers, from generated tokens to three non-overlapping parts of the context, following Yang et al. (2025b):

- **Needle**: tokens that correspond to the golden answer in the context. The "needle" tokens are generally distant from the generated tokens in the NIAH task.
- **Query**: tokens that correspond to the user question and the continuation prefix in the context. Thus, they are closest to the generated tokens.
- **Rest**: other tokens in the context.

We conduct our analysis on the NIAH dataset provided by

RULER (Hsieh et al.), where the context follows the format:

```
Rest ... Needle ... Rest ... Query
```

As shown in Table 4, for needle tokens that are distant yet critical for generation, our REPO method allocates substantially more attention mass than both the linear (i.e., RoPE) and constant (i.e., NoPE) position assignment strategies. Compared with REPO, the linear position assignment also exhibits a stronger locality bias, encouraging attention allocation to nearby query tokens. In addition, the constant position assignment, which treats all positions uniformly, produces an attention pattern with much lower variance across the three parts. These findings explain how our REPO method achieves performance gains on tasks involving noisy context, and also support our motivation based on Cognitive Load Theory (CLT), where the germane load (e.g., the attention mechanism) can better process the context information with context re-positioning.

### 5.2. Position Patterns Learned by REPO

To better understand the patterns learned by REPO, we analyze the characteristics of the assigned positions, first focusing on their ranges and then on their local patterns.

We first collect statistics on the distances between the maximum and minimum assigned positions for each attention head:

$$d^{k,h} = \max(\boldsymbol{z}^{k,h}) - \min(\boldsymbol{z}^{k,h}),$$

where $\boldsymbol{z}^{k,h} = \{z_1^{k,h}, z_2^{k,h}, \ldots, z_L^{k,h}\}$, $L$ is the number of tokens in input $\boldsymbol{x}$, and $k$ and $h$ represent the indices of the attention head and layer, respectively.

We compare these statistics on a general benchmark (2K-token context) and the RULER benchmark (4K-token context). As shown in Figure 3, we find that REPO assigns larger positional distances $d$ on longer context lengths, but the largest distance is still much smaller than the raw context length (i.e., 2K or 4K). This observation suggests that increasing the positional range to match the full input context length may not be necessary from the model's perspective. Furthermore, the distribution of distances is non-uniform, unlike the linear positional assignment in RoPE.

We hypothesize that assigning positions in a denser and non-linear continuous space contributes to improved performance on longer contexts in §4. Under the rotation framework of RoPE, the fixed training context length only covers a small portion of the period of low-frequency dimensions. Thus, the extrapolation for those dimensions would be more difficult, i.e., a more severe out-of-distribution issue. However, the learned denser and non-linear space can better leverage the small portion of the training context length to achieve better long-context performance.

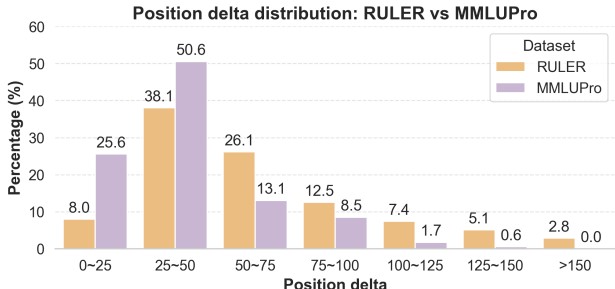

Figure 3. Statistics for the distances between maximum and minimum positions in each attention head of the LLM. The averaged and maximum number of tokens in the MMLUPro-Math benchmark are 1971 and 2512, while those for RULER-QA are 2995 and 3555, respectively.

We then analyze the patterns of the positions assigned by REPO. We split the positions $\boldsymbol{z}^{k,h}$ into non-overlapping chunks with $\Delta$ tokens $\{\boldsymbol{z}_{1:\Delta}^{k,h}, \boldsymbol{z}_{\Delta+1:2\Delta}^{k,h}, \ldots, \boldsymbol{z}_{L-\Delta:L}^{k,h}\}$ and define three pattern types[6]:

- **Constant**: We calculate the average position value $a$ in the chunk. If all positions lie between $[a - \epsilon, a + \epsilon]$, we conjecture that the positions are close to a constant, indicating the pattern of NoPE that assigns all positions to a constant position.
- **Mono**: If the positions in a chunk are monotonically increasing (i.e., $z_{i-1} < z_i < z_{i+1}$) or monotonically decreasing (i.e., $z_{i+1} < z_i < z_{i-1}$) for all $z_i$ in a chunk, we classify the pattern as *monotonic*, similar to the position assignment strategy used by conventional position encoding methods.
- **Hybrid**: All other patterns, e.g., a mixture of constant and monotonic patterns.

We empirically set $\Delta = 16$ and $\epsilon = 0.2$ to provide insights for the learned patterns[7]. As shown in Figure 4, we find that the **Mono** pattern is very rare (4% of all chunks), and the model prefers the constant patterns (22% of all chunks) over mono patterns. The dominating pattern of positions assigned by REPO is **Hybrid**, indicating that the position patterns beneficial for LLMs are different from those pre-defined in previous work (Vaswani et al., 2017; Su et al., 2024). This analysis is conducted on the RULER benchmark, but we find consistent observations on other benchmarks.

Besides the statistics of positions assigned by REPO, as shown in Appendix D, we also conduct a case study to visualize the positions across different layers and attention heads of the LLM. We find that the assigned positions can

---

[6]As shown in Figure 9, the three sub-figures are actually selected on purpose to demonstrate positions assigned by RePo that align with the three types of patterns.

[7]Setting different values for $\Delta$ and $\epsilon$ results in different plots, but the overall conclusion still holds.

*Table 5.* We use the default evaluation suites, including few-shot examples, prompts, and metrics, provided in `allenai/olmes` for evaluation. All evaluations are conducted within the training context length, i.e., 4096 tokens.

| Base | Method | ARC-C | ARC-E | BoolQ | CoQA | Drop | Hellaswag | MMLU-Pro | TriviaQA | AVG. | Δ vs RoPE |
|---|---|---|---|---|---|---|---|---|---|---|---|
| | RoPE | **47.99** | 75.25 | 72.12 | 56.87 | 37.90 | **70.68** | **13.77** | **54.98** | 53.70 | 0.00 |
| | NoPE | 44.05 | 73.64 | 66.70 | 44.52 | 33.22 | 65.68 | 10.70 | 43.43 | 47.74 | −5.96 |
| 1B | R2N1 | 47.30 | **75.68** | 73.04 | **59.31** | **38.48** | 69.26 | 13.41 | 54.61 | 53.88 | +0.18 |
| | N2R1 | 43.78 | 72.72 | 69.38 | 50.74 | 36.66 | 67.58 | 11.99 | 47.31 | 50.02 | −3.68 |
| | REPO | 47.61 | 74.87 | **73.58** | 57.44 | 38.17 | 70.08 | 13.52 | 54.56 | **53.73** | +0.03 |
| 7B | RoPE | **78.92** | **85.74** | 84.02 | **69.73** | **61.07** | **82.36** | **27.97** | **77.56** | **70.92** | 0.00 |
| | REPO | 78.46 | 85.61 | **84.32** | 68.59 | 59.59 | 81.92 | 27.15 | 76.80 | 70.30 | −0.62 |

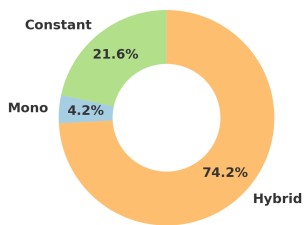

**Span-level Position Pattern Distribution**

Constant 21.6%
Mono 4.2%
Hybrid 74.2%

*Figure 4.* Statistics for the patterns of assigned positions. We split the context into non-overlapping chunks of 16 tokens. "Constant" means assigned positions are all close to a constant position, "Mono" means the positions are monotonically increasing or decreasing in the chunk, and "Hybrid" means all other patterns.

*Table 6.* Efficiency comparison. FLOPs are reported for training OLMo-2 1B model on 50B tokens. Inference time (second) per token is evaluated using vLLM.

| Method | FLOPs | Dec. Time / Token |
|---|---|---|
| RoPE | $3.84 \times 10^{20}$ | 0.0176 |
| REPO | $3.87 \times 10^{20}$ | 0.0182 |

comparable to that of the vanilla model.

## 6. Related Work

The self-attention mechanism in Transformers (Vaswani et al., 2017) is inherently permutation-invariant, lacking an intrinsic understanding of input token order. To address the issue, a position encoding module is used in most Transformer-based models to map the position indices of input tokens into biases or embeddings that can be integrated into the model.

Bias-based position encoding methods, such as ALiBi (Press et al., 2021), KERPLE (Chi et al., 2022), and T5 (Raffel et al., 2020), integrate a distance bias directly into the attention logits to control the attention field of perception. Most position encoding methods, however, learn embeddings for position indices or distances. The absolute position embedding, which is added to the hidden states, has been widely used in Transformer-based models with small sequence lengths (Vaswani et al., 2017; Radford et al., 2019; Devlin et al., 2019). In the era of large language models (LLMs), the RoPE method (Su et al., 2024), which rotates queries and keys for calculating attention scores based on position distance, has become the *de facto* choice for modern LLM architectures. Many subsequent variants have been proposed to understand and improve the context extrapolation performance of RoPE (Peng et al., 2024; Chen et al., 2023; 2024; 2025; Oka et al., 2025; 2026b;a; Li et al., 2026), and exploring the combination with NoPE (Barbero et al., 2025; Yang et al., 2025b; Gelberg et al., 2025) at different levels. Our work is orthogonal to these enhanced RoPE methods.

In contrast to previous literature, our work focuses on position prediction for tokens within a given context, prior to the

capture the intrinsic structure of the input context, such as the segmentation of few-shot examples, which aligns with our CLT-based motivation.

### 5.3. Performance on General Tasks

As shown in Table 5, along with the noticeable performance gain in previous experiments, our REPO method still achieves performance comparable to the RoPE method on extensive general benchmarks. This occurs even though changing from linear position assignment to REPO causes an inconsistency between pre-training and continued training. This observation indicates that our REPO, learned from general data, generalizes well to diverse types of data, even when questions in general benchmarks are typically short and precise, requiring almost no context reorganization.

### 5.4. Efficiency

Table 6 compares the FLOPs and inference time between the vanilla model with the RoPE method and REPO. We observe that the REPO method is very lightweight, introducing only a 0.9% increase in parameters while providing performance gains across many evaluation dimensions. When running inference for RULER benchmark (Hsieh et al.) within training context length, the time cost of REPO is

position encoding process. The most relevant work to ours is COPE (Golovneva et al., 2024), which learns an attention-logits-based gate module to predict a non-decreasing sequence for positions, aiming to segment high-level abstractions, such as sentences. However, the COPE method uses attention logits for gating, which incurs high time and memory costs due to the computation of $[B, L, L]$ tensors. It is also incompatible with other position encoding methods, like RoPE, and flash attention (Dao, 2024), limiting its scalability. More data-dependent position encoding methods are further designed from the perspective of linear Transformers (Lin et al., 2025; Yang et al., 2025c; Movahedi et al., 2025). In contrast, our work focuses on re-organizing tokens within a context using the lightweight REPO that is compatible with most positional encoding methods and can be applied to standard pre-trained LLMs without training from scratch.

There also exist works that re-organize the context in a model-agnostic way. (Peysakhovich & Lerer, 2023) demonstrate that manually putting the documents with highest attention scores to the end of context achieves more robust performance on long-context question answering. Recently, Zhang et al. (2025) also highlighted the importance of input context and proposed ACE, which iteratively evolves the context in an agentic manner. Our work aims to enhance Transformer architecture with context re-positioning, which is orthogonal to this line.

## 7. Conclusion

In this paper, we reduce the cost raise by the oversimplified organization of context, i.e., *extraneous cognitive load*, by substituting the rigid linear position encoding in LLMs with context Re-Positioning (REPO). The proposed REPO is a novel, lightweight differentiable module ($f_\phi$) that empowers models to learn positions that capture the structure and dependencies in a context. Through continual pre-training of OLMo-2 models, our extensive experiments demonstrated that REPO significantly outperforms strong baselines on tasks requiring specific contextual dependencies, showing substantial gains on noisy-context, structured data, and long-context extrapolation benchmarks, while maintaining performance on general short-context tasks. Our analysis revealed that REPO also brings notable technical advantages, as it mitigates locality bias of traditional position assignment strategies by allocating more attention to distant but relevant "needle" tokens, and learns adaptive position patterns in a more non-linear and denser continues space. By enabling models to actively re-position their input context, REPO opens a new direction for flexible context management driven by innovations in LLM architecture.

## Impact Statement

This work improves how large language models (LLMs) organize and attend to contextual information, particularly in long or noisy inputs. By enabling more effective use of relevant context, the proposed method can improve reliability and efficiency in downstream applications such as long-document understanding, retrieval-augmented generation, and agentic systems.

The method does not introduce new data sources or supervision and operates within existing model architectures. While improved contextual reasoning may amplify both beneficial and harmful uses of language models, this work does not inherently introduce new ethical risks beyond those already associated with LLMs, and highlights context organization as an important direction for improving robustness and interpretability.

## Acknowledgment

The authors thank Yoav Gelberg and Yingtao Tian for insightful discussions regarding the experiments, as well as Yujin Tang, Qi Sun, Makoto Shing, Lemao Liu, Leyang Cui, Yahui Liu, Tian Lan, and Minghao Wu for their feedback on the manuscript; and the anonymous reviewers and chairs for their time and constructive comments.

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

# A. What's the Real Difference between Conventional PEs, NoPE, and RePo?

In the background section (§2), we use ROPE as a representative example to illustrate how conventional positional encoding methods rely on a strict linear pattern to assign positional information to tokens in the context.

Recently, researchers have found that the causal mask in the attention mechanism enables LLMs to implicitly learn positional information, and that removing explicit positional encoding can even achieve superior performance on structured data and long-context tasks. This approach is referred to as the NoPE method (Kazemnejad et al., 2023; Yang et al., 2025b; Wang et al., 2024a; Barbero et al., 2025). We argue that the attention score of NoPE can be reformulated within the RoPE framework by assigning a constant positional value $a$:

$$
\begin{aligned}
\mathbf{A}_{i,j}^{\text{NoPE}} &= \boldsymbol{q}_i^\top \boldsymbol{k}_j \\
&= \boldsymbol{q}i^\top g\theta(0)\boldsymbol{k}_j \\
&= \boldsymbol{q}i^\top g\theta(a - a)\boldsymbol{k}_j,
\end{aligned}
\tag{8}
$$

where $a$ denotes a uniform position value for all tokens, yielding a constant rotation matrix $g_\theta(0)$. Thus, under this reformulation, the key difference between RoPE and NoPE lies solely in how positions are assigned.

In addition, LLMs with interpolated NoPE and RoPE layers (Yang et al., 2025b; Barbero et al., 2025) have been widely used architectures, which can be explained as hybrid position assignment strategies. In contrast to previous works that empirically configure the hybrid system with hyper-parameters, our REPO shows higher expressiveness, as it can dynamically determine whether to adopt the conventional linear, NoPE-like constant, or hybrid position assignment for tokens in a given context. A comparison among the three approaches is summarized in Table 7. As explained in §2, when $z_j = z_i$, REPO effectively reduces to the NoPE pattern with the constant $z_j = z_i = a$. In contrast, if $z_j > z_i$ for $j > i$, it indicates that REPO adopts positional relationship similar to the conventional linear style, e.g., the strategy used in RoPE. In our experiments and analyses, we will demonstrate

*Table 7.* Comparison between different methods. In RoPE-like methods, $g_\theta$ generates a rotation matrix based on a distance. The $j - i$ is the distance between $x_j$ and $x_i$, $g_\theta(0)$ is a constant rotation, and $z_j$ and $z_i$ are predicted positions by $f_\phi$ (Eq. 7).

| Method | Attention Score |
|---|---|
| Linear (e.g., ROPE) | $\boldsymbol{q}_i^\top g_\theta(j - i)\boldsymbol{k}_j$ |
| Constant (e.g.,NOPE) | $\boldsymbol{q}_i^\top g_\theta(0)\boldsymbol{k}_j$ |
| REPO | $\boldsymbol{q}_i^\top g_\theta(z_j - z_i)\boldsymbol{k}_j$ |

that an LLM may dynamically select between constant and linear position assignments, or hybridize them with REPO module. Notably, although we use RoPE for the comparison, linear position assignment is widely adopted in conventional positional encoding methods (Vaswani et al., 2017; Gehring et al., 2017; Press et al., 2021; Li et al., 2025), and our findings can be readily extended to these approaches.

# B. Details of Experiments

## B.1. Noisy Context

We use the following task ids in `olmes` for the evaluation in Table 1: `ruler_niah_s_1__4096::std`, `ruler_niah_mk_1__4096::std`, `ruler_niah_mv__4096::std`, `ruler_niah_mq__4096::std`

## B.2. Extrapolation

We use the following hyper-parameters to extend the context:

1. 8K Tokens: {"rope_type": "yarn", "factor": 2.0, "original_max_position_embeddings": 4096}

2. 16K Tokens: {"rope_type": "yarn", "factor": 4.0, "original_max_position_embeddings": 4096}

We use the setting for "16K Tokens" for all the experiments on LongBench (Table 3).

## B.3. General Tasks

We use the following task ids in `olmes` for the evaluation in Table 5: `arc_challenge:rc::large`, `arc_easy:rc::olmes`, `boolq:rc::large`, `coqa::large`, `drop::large`, `hellaswag:rc::large`, `mmlu_pro:cot::none`, `triviaqa::olmes`

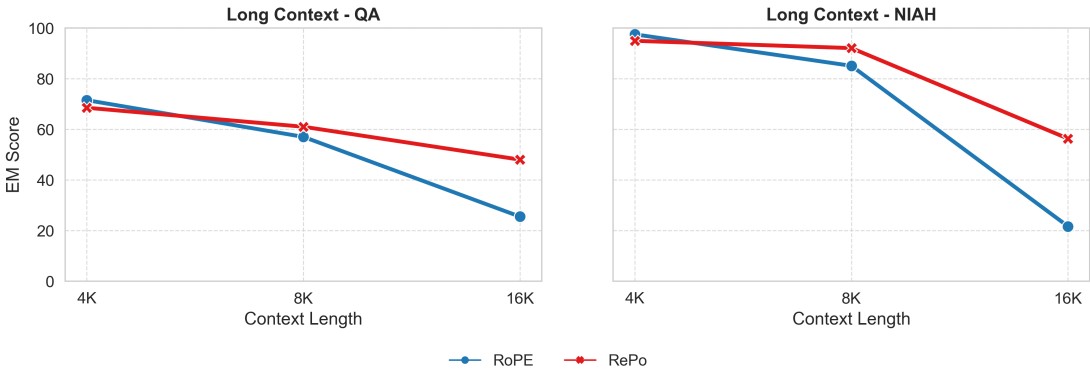

*Figure 6.* Long-context evaluation on OLMo-2 7B model. We use the data in RULER (Hsieh et al.) benchmark and apply YaRN (Peng et al., 2024) for all RoPE layers to extend the context.

### B.4. Additional Results on Long Context

Results for the context extrapolation on OLMo-2 7B model are shown in Figure 6.

### B.5. Ablation Study

As shown in Figure 5, we evaluate the sensitivity of model performance to the starting layer of REPO on OLMo-2 1B model, where $l = 5$ indicates that REPO is applied beginning from the 5th layer of the LLM, while the vanilla ROPE is used for the lower layers. We conduct experiments on two subtasks, NIAH and MMLUPro. The results show that overall performance is robust to this hyperparameter. However, increasing $l$ slightly improves performance on general benchmarks while negatively affecting performance on NIAH. The results are consistent on other evaluation benchmarks.

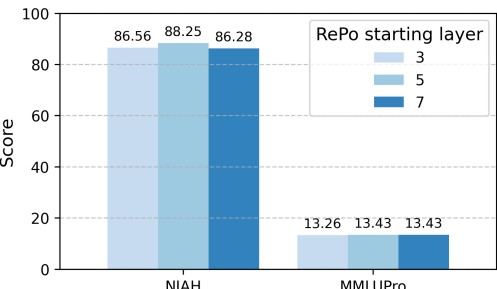

*Figure 5.* Sensitivity to the starting layer of REPO (i.e. $l = 3, 5, 7$). We validate on the NIAH subtask of RULER benchmark and MMLUPro of general benchmarks.

## C. Preliminary Experiments

In this experiment, we train a small-scale language model on a purposefully selected synthetic task, namely text reversal, to determine whether REPO can re-position tokens in the context.

In the text reversal task, a model is prompted to generate a given sequence of tokens $\boldsymbol{x} = \{x_1, x_2, \ldots, x_L\}$ in a reversed order $\boldsymbol{x}' = \{x_L, x_{L-1}, \ldots, x_1\}$. *Locality bias* does not apply here because the distance between a generated token and its corresponding dependent input token grows linearly as the generation proceeds. It is interesting to investigate whether the REPO method can learn beneficial re-positioning patterns from the task.

### C.1. Setup

We use the data and train/dev/test splits provided in Kazemnejad et al. (2023) for the text reversal task. The sequence lengths $L$ in training are between $[2, 20]$, while we use the sequence lengths between $[2, 30]$ for evaluation. The input sequence $\boldsymbol{x}$ is constructed from a fixed set of subwords that are shared across the three datasets, without regard to grammatical or semantic structure (Kazemnejad et al., 2023).

We train a GPT-2 model with 4 layers for this task. We use NoPE, RoPE, and REPO[8] methods to train the model. For the REPO method, we shared the parameters of $f_\phi$ for all the attention heads in each layer. All training hyper-parameters are set as in Kazemnejad et al. (2023).

---

[8]Notably, REPO functions solely as a position prediction module. For brevity, however, mentioning REPO implies the use of RoPE encoding together in this work.

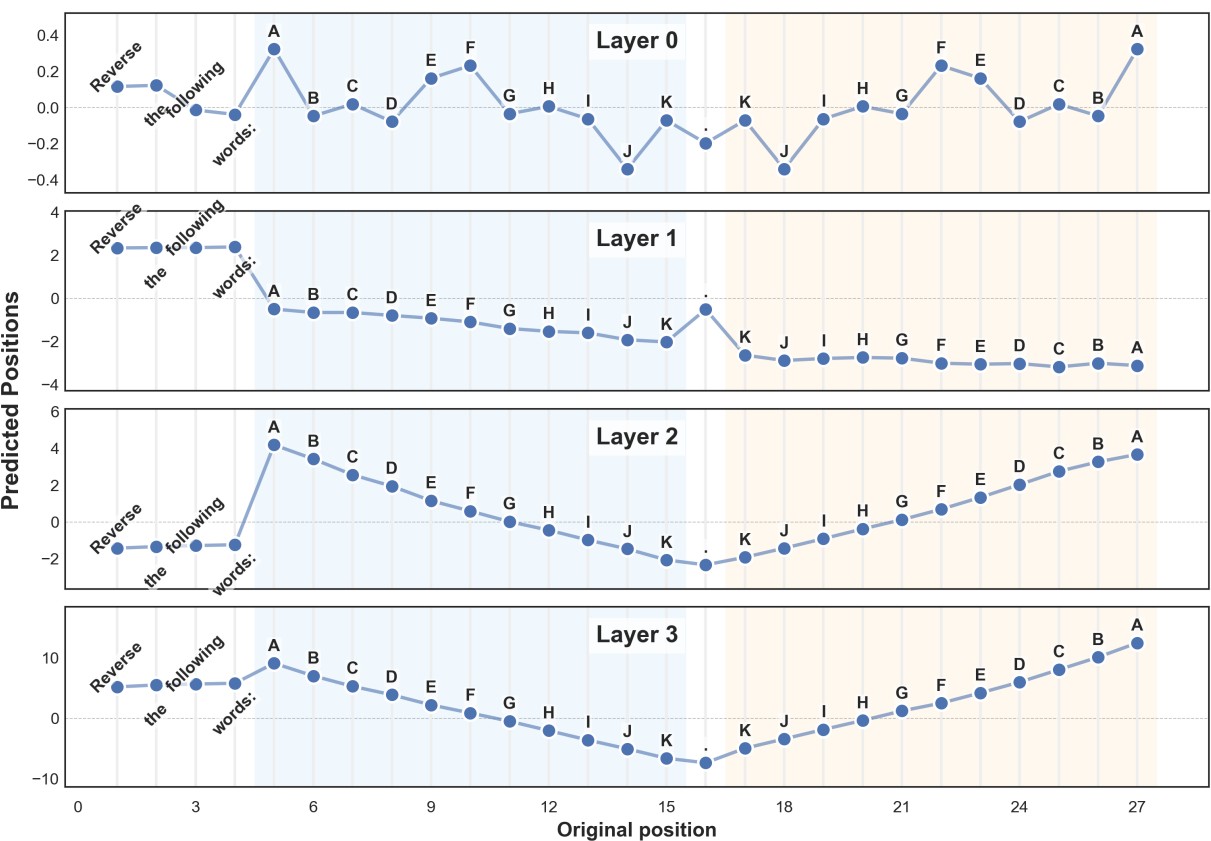

*Figure 7.* Visualization of predicted positions from a 4-layer GPT-2 model in the reversal task. The area with blue background color indicates input context, while the orange region is the generated sequence. We use A-K to replace the real tokens to save space for illustration. The x and y-axis represent the input order and predicted position $z$ of a token, respectively.

## C.2. Findings

As shown in Figure 8, due to the simplicity of the text reversal task, all the models achieve nearly perfect performance on short in-domain sequences ($L \leq 20$). However, when testing on examples with longer-range dependencies, i.e. $L > 20$, our REPO method demonstrates superior performance compared to both NoPE and RoPE.

We further investigate the re-positioning patterns learned by REPO that contribute to this performance gain. As illustrated in Figure 7, we visualize the predicted positions across different layers of the trained model and observe several intriguing patterns. The overall distribution of predicted positions is remarkably distinct from the pre-defined positional indices (e.g., 1 to 27). Specifically, we observe a mirror effect in layers 0, 2, and 3, where pairs of reversal tokens are assigned the same position indices. Additionally, we identify hybrid positional patterns across

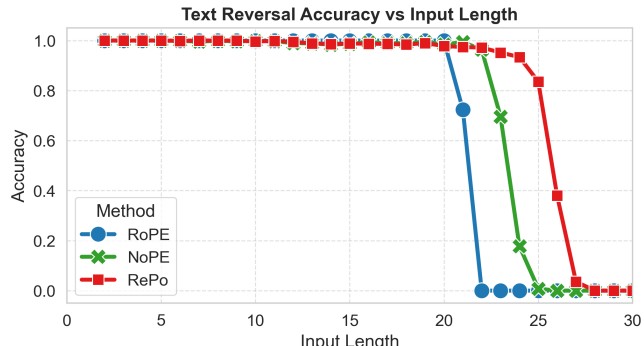

*Figure 8.* Performance on the text reversal task. We report the accuracy on all lengths of input sequences.

different parts of the context. For example, from layers 1 to 3, the model adopts a NoPE-like pattern for the opening tokens, assigining nearly identical position indices to tokens in the phrase "Reverse the following words:", while exhibiting patterns with mirror symmetry for the input and output sequences. Notably, we did not introduce any inductive bias for this task; all patterns emerged in a purely data-driven manner. These intriguing patterns motivate us to further investigate REPO on more general datasets.

## D. Case Study

As shown in Figure 9, we visualize the positions assigned by REPO when testing on the MMLUPro benchmark (Wang et al., 2024b) with few-shot examples. We observe that REPO learns distinct patterns across different layers and attention heads. Interestingly, as shown in Figure 9b, the patterns of assigned positions roughly align with the semantic segmentation of the few-shot examples, demonstrating that REPO is capable of capturing the structure of the input context. Additionally, we find that some positions assigned by REPO are negative values, as shown in Figure 9c. Those negative positions can be interpreted as rotations in a reversed direction under the RoPE framework. There also exist some outlier positions in the figures. Upon inspection, we find that they correspond to non-informative punctuation marks and function words, such as "." and "such."

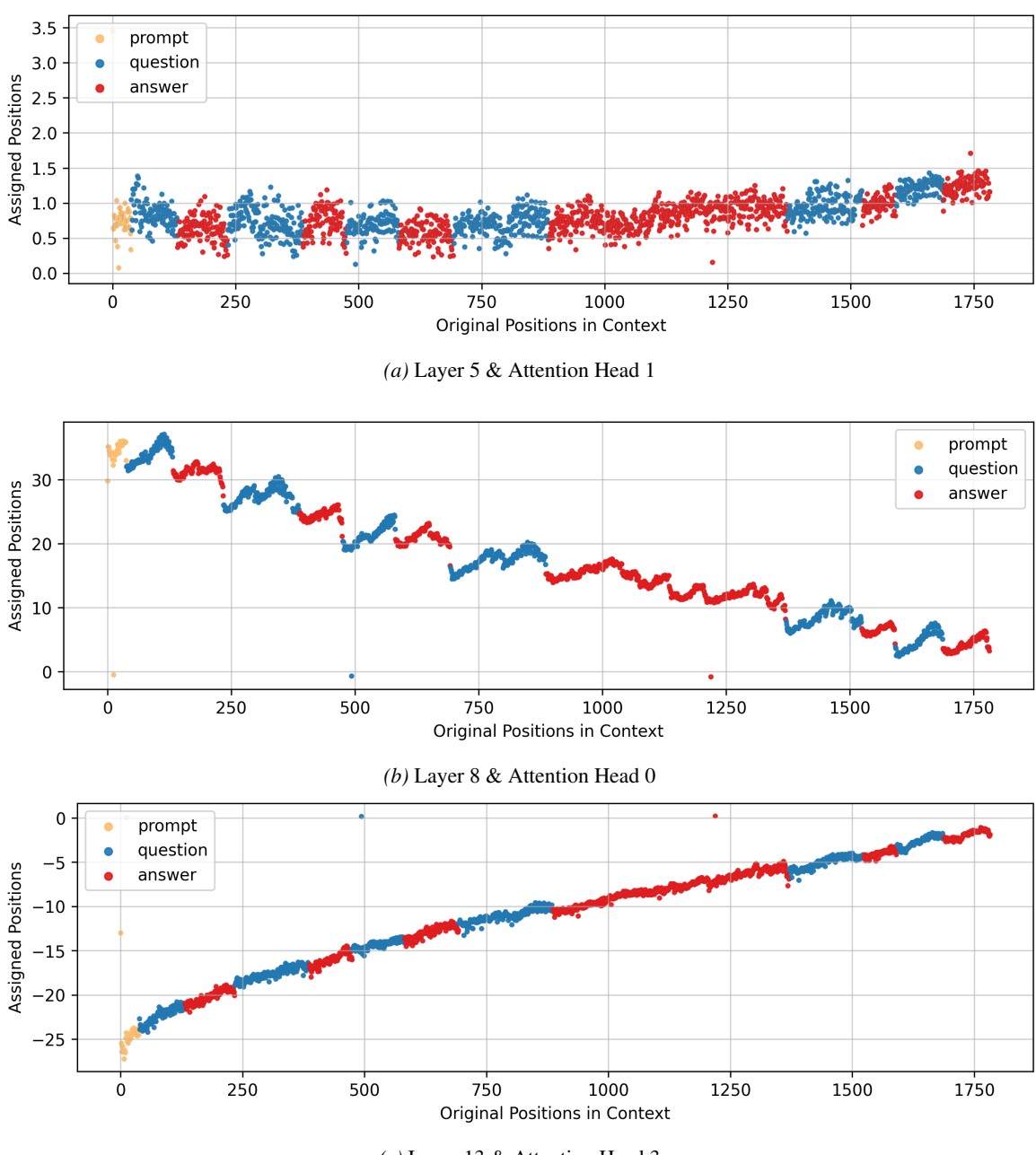

*(a)* Layer 5 & Attention Head 1

*(b)* Layer 8 & Attention Head 0

*(c)* Layer 13 & Attention Head 3

*Figure 9.* Visualization of positions assigned by REPO. The REPO is continuously trained on general data. The visualization data is from MMLUPro with few-shot examples. Symbols in orange belong to the prompt, while symbols in blue and red represent questions and answers in few-shot examples.

