# OpenReview forum: "RePo: Language Models with Context Re-Positioning"
_ICML.cc/2026/Conference — ICML 2026 regular_

### Official Review · Reviewer_eWoC · 2026-03-08

**Soundness:** 3
**Presentation:** 2
**Significance:** 3
**Originality:** 3
**Overall Recommendation:** 4
**Confidence:** 4

**Summary:**

This paper explores the limitations imposed by fixed positional encodings in LLMs and proposes REPO motivated by CLT. The authors argue that traditional linear or constant position indexing unnecessarily consumes the model's capacity to process core information. To address this, the paper introduces a lightweight differentiable module that dynamically assigns position values in a continuous non-linear space based on the hidden states and contextual dependencies of the input tokens. This approach aims to break the constraints of absolute sequencing and optimize the model's attention allocation in long texts and complex contexts.

**Compliance With Llm Reviewing Policy:**

Affirmed.

**Final Justification:**

My questions have been adequately addressed. I will maintain my score, as this remains a technically solid and practically valuable contribution.

**Key Questions For Authors:**

1. Could you provide concrete benchmark data regarding efficiency? For instance, under identical hardware conditions, what is the exact percentage degradation in latency and throughput during the inference phase (both Prefill and Decode stages) for the 7B model equipped with REPO compared to the standard architecture?

2. Why was the decision made to apply this mechanism specifically to the "latter 2/3 deep layers" of the model? Are there corresponding ablation studies to support this choice? How would the trade-off between performance and efficiency look if it were applied only to the final few layers, or via an alternating layers strategy?

3. Dynamically altering positions often introduces additional training noise. Does REPO lead to any performance degradation on standard short-text reasoning tasks (such as standard MMLU or GSM8K) that do not necessitate long-range dependencies?

**Limitations:**

yes

**Strengths And Weaknesses:**

Strengths：

1. Introducing CLT from cognitive psychology to explain the inefficiency of standard linear positional encoding provides a highly intuitive and cross-disciplinary perspective for understanding the flaws in LLM attention mechanisms.

2. REPO translates discrete position indices into content-based continuous real values. Notably, it only modifies relative positions during the attention computation phase without altering the actual physical autoregressive sequence. This elegant compromise perfectly circumvents the catastrophic computational overhead of recomputing the KV Cache, giving it high practical deployment value.

3. Through attention matrix analysis, the paper clearly demonstrates that REPO breaks the "Locality Bias" of standard RoPE, genuinely achieving attention allocation based on "relevance rather than physical distance." This provides solid evidence for the model's enhanced capabilities in handling long contexts and structured data.

Weaknesses：

1.  While CLT serves as an appealing motivation, from a rigorous machine learning perspective, REPO is essentially learning a "content-dependent dynamic relative position bias." I recommend exercising restraint in the narrative, framing it as a heuristic analogy to prevent readers from mistakenly believing the model genuinely possesses a working memory structure in the human psychological sense.

2. The empirical data shows significant gains for REPO on the 1B model, but the absolute improvement shrinks to a marginal margin (e.g., 0.6 points) on certain tasks for the 7B model, such as the NIAH test. This raises concerns about whether this method remains effective on much larger-scale models (e.g., 70B+), where native attention heads might already possess highly robust information routing capabilities.

3. Although the paper claims the design is highly efficient, the introduction of the additional differentiable module $f_\phi$ (applied to the latter 2/3 layers) inevitably increases FLOPs and latency. Currently, there is a lack of concrete tables comparing Training Throughput (tokens/s) and Inference Latency (ms) between the baseline and REPO models.

---

> ### Author Rebuttal · Authors · 2026-03-30
>
> > **Q1: While CLT serves as an appealing motivation, from a rigorous machine learning perspective, REPO is essentially learning a "content-dependent dynamic relative position bias." I recommend exercising restraint in the narrative, framing it as a heuristic analogy to prevent readers from mistakenly believing the model genuinely possesses a working memory structure in the human psychological sense.**
>
> Thanks for your helpful feedback. We will restructure the introduction part to avoid misunderstanding by readers.  In addition, our analysis experiments, e.g., the improved attention allocation by RePo in Sec 5.1, are designed to connect with CLT. We will strengthen this connection and highlight that RePo is learning “content-dependent dynamic relative position bias.” in the revised paper.
>
> > **Q2: The empirical data shows significant gains for REPO on the 1B model, but the absolute improvement shrinks to a marginal margin (e.g., 0.6 points) on certain tasks for the 7B model, such as the NIAH test. This raises concerns about whether this method remains effective on much larger-scale models (e.g., 70B+), where native attention heads might already possess highly robust information routing capabilities.**
>
> Thanks for your concern. We agree that RePo only outperforms RoPE by 0.6 EM scores on NIAH. This is partially because the performance of the base models is already very high on NIAH tasks, i.e., 97.3 EM scores. Thus, improving 0.6 EM scores is non-trivial, i.e., 22% reduction of the remaining errors. It is worth noting that the performance gains of RePo on the 7B model are also compelling on structured data (Tab. 2) and long-context (Tab. 3).
>
> > **Q3: Although the paper claims the design is highly efficient, the introduction of the additional differentiable module  (applied to the latter 2/3 layers) inevitably increases FLOPs and latency. Currently, there is a lack of concrete tables comparing Training Throughput (tokens/s) and Inference Latency (ms) between the baseline and REPO models.**
>
> We appreciate your suggestion for the efficiency comparison. In Table 6 (Sec 5.4) of the main paper, we compare the FLOPs and decoding time per token between RePo and the standard RoPE method. We use the RULER benchmark for this analysis. As shown in the results, the lightweight RePo module adds marginal overhead to the base model.
>
> > **Q4: Could you provide concrete benchmark data regarding efficiency? For instance, under identical hardware conditions, what is the exact percentage degradation in latency and throughput during the inference phase (both Prefill and Decode stages) for the 7B model equipped with REPO compared to the standard architecture?**
>
> Please refer to the response in Q3 for more details.
>
> > **Q5: Why was the decision made to apply this mechanism specifically to the "latter 2/3 deep layers" of the model? Are there corresponding ablation studies ... alternating layers strategy?**
>
> Thank you for the interesting question. The intuition behind the layer threshold is discussed in Section 4.2 (Training & Efficiency) between lines 149 and 180. This design choice is motivated by previous papers that lower layers of LLMs primarily capture surface-level features which depend more on local information, thus re-organization of context at lower layers may be burdensome to the LLM. Our ablation study for this hyper-parameter is in Appendix B.5, which shows that our model is relatively robust to this parameter.
>
> In our preliminary study, we tried many strategies to apply RePo. First, we tried to apply RePo only to very few layers, and the performance of RePo after training becomes very close to RoPE — that is, the context repositioning module doesn’t impact the system significantly. However, applying to the last layer achieves almost the same efficiency as RoPE. Second, we also tried the reverse strategy, i.e., applying RePo to the former ⅓ layers and standard RoPE to the latter ⅔ layers: The training becomes unstable under this setting.  We will discuss those explorations in the revised paper.
>
>
> > **Q6: Dynamically altering positions often introduces additional training noise. Does REPO lead to any performance degradation on standard short-text reasoning tasks ... long-range dependencies?**
>
> Thank you for the good question. We actually conducted extensive experiments to better understand the impact of RePo and demonstrate its general performance. As shown in Sec. 5.3, the overall performance of RePo on short-context tasks, such as MMLU-Pro and ARC,  is comparable with the RoPE method, with significant performance gains on long-context and structured data. This observation indicates that our RePo, learned from general data, generalizes well to diverse types of data, even when questions in general benchmarks are typically short and precise, requiring almost no context reorganization.

---

> > ### Author Rebuttal · Reviewer_eWoC · 2026-04-03
> >
> > Thank you for the detailed rebuttal. My questions have been adequately addressed. I will maintain my score, as this remains a technically solid and practically valuable contribution.

---

### Official Review · Reviewer_GzVy · 2026-03-12

**Soundness:** 3
**Presentation:** 3
**Significance:** 3
**Originality:** 3
**Overall Recommendation:** 5
**Confidence:** 3

**Summary:**

This paper challenges the limitations of rigid, linear position encoding in LLMs by leveraging Cognitive Load Theory (CLT) to argue that fixed structures introduce "extraneous load," which impedes effective attention allocation. To address this, the authors introduce REPO, a lightweight and differentiable module ($f_{\phi}$) that dynamically maps token positions into a continuous, non-linear space based on their hidden states. By evaluating REPO through the continued pre-training of OLMo-2 1B and 7B models on 50B tokens, the results reveal significant performance improvements in noisy contexts, structured tables, and long-context extrapolation up to 16K, all while maintaining robust results on standard short-context benchmarks.

**Compliance With Llm Reviewing Policy:**

Affirmed.

**Key Questions For Authors:**

See weakness

**Strengths And Weaknesses:**

**Strength**
1. Use Cognitive Load Theory to provide a novel motivation for moving beyond linear position assignments.

2. The REPO module is highly lightweight, adding only 0.9% to the total parameter count and introducing negligible inference latency.

3. The gains on HybridQA and LongBench are substantial, particularly given that these improvements were achieved through a relatively small continued training budget of 50B tokens.

**Weakness**

1. The CLT motivation is compelling, but I didn't see a direct connection between CLT theory and PE, making it feel like a story but not a convincing theory. The paper would benefit from a more quantitative link between "cognitive load" and specific Transformer metrics.

2. I notice that the author mentioned that COPE is a similar method to this work, but there isn't a direct comparison between COPE and REPO.

3. The evaluation is limited to the OLMo-2 family. Testing on a broader range of architectures would further validate the universality of the mechanism. I understand this costs many computing resources, so this is only a minor concern.

---

> ### Author Rebuttal · Authors · 2026-03-30
>
> > **Q1: The CLT motivation is compelling, but I didn't see a direct connection between CLT theory and PE, making it feel like a story but not a convincing theory. The paper would benefit from a more quantitative link between "cognitive load" and specific Transformer metrics.**
>
> Thanks for your helpful suggestion. We will revise the paper to make the CLT part more solid and easier to understand for readers. Our analysis experiments, e.g., the improved attention allocation by RePo in Sec 5.1, are designed to connect with CLT. We will strengthen this connection more clearly in the revised paper. Please let us know if you have any suggestions for more appropriate Transformer metrics, and we are more than happy to integrate them into the analysis part.
>
> > **Q2: I notice that the author mentioned that COPE is a similar method to this work, but there isn't a direct comparison between COPE and REPO.**
>
> Thank you for your suggestion.  We actually have spent a lot of effort on the COPE baseline. Firstly, the CoPE method requires the exact attention score for computing the position value, which is unsupported by Flash Attention. Training with a naive attention implementation is a disaster for LLMs with more than 1B parameters, easily leading to OOM. To fix the OOM issue, we set batch size to 1 and enable the activation checkpointing to save the memory usage on H100. It takes around 2 weeks to fine-tune the 1B model with CoPE on midsize training data.
> Secondly, unlike RePo which works compatibly with almost all the position encoding methods, CoPE is actually another position encoding method. We train CoPE under a setup similar to other baselines to ensure a fair comparison. However, directly fine-tuning CoPE on the pre-trained 1B model causes more severe loss spikes, and the final performance of CoPE is worse than all the baselines. We adopt the official code from the CoPE paper for the implementation. We will release the code for CoPE and its configuration together with our codebase so that others can reproduce our results.
>
>
> > **Q3: The evaluation is limited to the OLMo-2 family. Testing on a broader range of architectures would further validate the universality of the mechanism. I understand this costs many computing resources, so this is only a minor concern.**
>
>
> We sincerely thank the reviewer for the understanding regarding the computational costs. We completely agree that evaluating RePo across a broader range of LLMs would further strengthen this work. We will explore the training on other model families and models with larger size in future works.

---

> > ### Author Rebuttal · Reviewer_GzVy · 2026-04-02
> >
> > You use CLT as a core motivation for your paper, yet in the end you only rely on a result-driven experiment to argue that "your method is indeed related to CLT." This logic is clearly flawed and fails to explain why CLT is being used as a motivation in the first place. This renders your paper's reasoning susceptible to circular argumentation.
> >
> > Please give detailed plan about "We will revise the paper to make the CLT part more solid and easier to understand for readers." Otherwise I cannot be convinced that why CLT becomes the motivation of your method.

---

> > > ### Author Response · Authors · 2026-04-04
> > >
> > > Thank you for further pointing out the problematic positioning of CLT in the paper.
> > >
> > > Overall, we are excited about bridging the gap between LLMs and CLT, as this intersection holds significant potential for LLM architecture and agent design. We acknowledge, however, that this concept remains under-explored in the context of LLMs, lacking standard testbeds to directly measure improvements based on CLT. To the best of our knowledge, attention allocation currently serves as one of the most viable proxies for this measurement. Nevertheless, we agree that the reviewers' suggestions regarding the CLT framing are highly constructive. Consequently, we have decided to restructure our paper to better reflect this feedback.
> > >
> > > After careful reflection and considering the suggestions from all reviewers, we realize that, instead of being framed as the initial motivation of the entire RePO method, CLT should be introduced as a theoretical lens that helps one understand why it is beneficial to move the burden of organizing text structure from attention layers to a specific re-positioning module.
> > >
> > > For the detailed revision plan, we will implement the following.
> > > - Avoid framing CLT as a “core motivation” anywhere in the paper.
> > > - Emphasize CLT as a post-hoc analytic tool, adjusting the causality between CLT and the RePO method.
> > > - Bridging CLT with the analysis results more closely by drawing an analogy between germane load and the attention mass on Needle tokens in Section 5.1.
> > >
> > > While we will apply the revision to the entire paper, we would like to show a couple of concrete examples of the revision.
> > >
> > > **Abstract**
> > >
> > > Before:
> > > > Drawing on Cognitive Load Theory (CLT), we argue that this uninformative structure increases extraneous cognitive load, consuming finite working memory capacity that should be allocated to deep reasoning and attention allocation. To address this, we propose RePO, a novel mechanism that reduces extraneous load via context re-positioning.
> > >
> > > After:
> > > > The rigid position information poses the full burden of organizing the input structure to attention layers, thus reducing the amount of attention that could be allocated for more critical information. To address this, we propose RePO, a novel mechanism that alleviates the burden for attention layers via context re-positioning.
> > >
> > > **Instruction paragraph 3-4**
> > >
> > > Before:
> > >
> > > > Although fixed position assignments have become the de facto standard, they deviate from how human working memory processes information. According to Cognitive Load Theory (CLT), the capacity of working memory during problem solving can be consumed by costs arising from how information is organized and presented, referred to as extraneous load (Sweller, 1994; Paas et al., 2003). CLT studies suggest ... reorganization can be reallocated to deeper reasoning processes associated with the germane load, thereby improving problem-solving performance (Sweller, 1994).
> > > >
> > > >
> > > > However, the critical ability to reorganize and restructure contextual information (Vaswani et al., 20017; Yang et al., 2025a; Dubey et al., 2024) ... highly diluted contexts (Hsieh et al.), exhibit notable performance degradation, mirroring the effects predicted by CLT under high extraneous load.
> > >
> > > After:
> > >
> > > > Although fixed position assignments have become the de facto standard, they deviate from how human working memory processes information. Studies on human learning show that the presence of structural information facilitates text processing (Hyönä and Lorch, 2004; Schneider et al., 2018). However, the capability of turning linearized text into a structured representation is absent from the architectural design of modern LLMs. As a result, the attention mechanism is fully responsible for understanding the underlying language structure of linearized texts. This burden impairs the computational allocation for contextual reasoning, thus harming model performance in tasks that require strong long-range or fine-grained contextual dependencies, e.g., needle-in-a-haystack (NIAH) problems (Kamradt, 2023) or question answering under highly diluted contexts (Hsieh et al.). This can be additionally interpreted in terms of Cognitive Load Theory (CLT), which states that the capacity of working memory is allocated to intrinsic load (the inherent difficulty of the task), extraneous load (how information is organized and presented), and germane load (the learning process). While the task difficulty is fixed, an attention layer has to assign its limited capacity to understanding the structure of an input sequence and processing the input. The performance of the latter degrades when the former consumes too much capacity.
> > >
> > > **Please let us know if you have any other questions or concerns. We are more than happy to discuss.**
> > >
> > > **Reference**
> > >
> > > [Hyönä and Lorch, 2004] Effects of topic headings on text processing: evidence from adult readers’ eye fixation patterns
> > >
> > > [Schneider et al., 2018] A meta-analysis of how signaling affects learning with media

---

### Official Review · Reviewer_uefV · 2026-03-12

**Soundness:** 4
**Presentation:** 4
**Significance:** 3
**Originality:** 4
**Overall Recommendation:** 5
**Confidence:** 4

**Summary:**

The paper addresses the limitation of fixed, linear positional encodings (like RoPE) in Large Language Models (LLMs), which the authors argue imposes a rigid contextual structure that increases "extraneous cognitive load" according to Cognitive Load Theory (CLT). To solve this, the authors propose REPO, a lightweight, differentiable module ($f_{\phi}$) that dynamically assigns continuous-valued positions to tokens based on their hidden states. Unlike standard approaches that use a fixed $0 \dots L-1$ order, REPO learns to re-position tokens in a non-linear space to better capture contextual dependencies. Through continual pre-training on OLMo-2 1B and 7B models, the authors demonstrate significant performance gains in tasks requiring long-range dependencies, structured data (e.g., tables), and long-context extrapolation, while maintaining competitive performance on general short-context benchmarks.

**Compliance With Llm Reviewing Policy:**

Affirmed.

**Final Justification:**

The authors have addressed my concerns, and I will keep my original score.

**Key Questions For Authors:**

The experiments are conducted on 1B and 7B models. Do the authors expect the benefits of REPO to scale similarly to larger models (e.g., >30B), where positional inductive biases may interact differently with attention patterns?

How sensitive is the performance to the $1/8$ hidden size reduction ($d_p < d$)? Would a larger representation allow for more complex structural capture, or is position truly a low-dimensional signal?

Could you provide a direct latency/FLOPs comparison with COPE to quantify the "high time and memory costs" mentioned in your related work?

**Limitations:**

The authors adequately discuss the technical limitations and broader impacts, noting that the method is orthogonal to other context-reorganization strategies and does not introduce new data-related ethical risks.

**Strengths And Weaknesses:**

#### **Soundness**

- **Strengths:**
  - The connection to Cognitive Load Theory provides an intuitive motivation for exploring dynamic context reorganization in language models.
  - The method is evaluated on two model scales (1B and 7B) using a fully open-source backbone (OLMo-2) to avoid data contamination issues, which strengthens the validity of the results.
  - The paper provides a comprehensive suite of experiments, including Needle-In-A-Haystack (NIAH), structured data (HybridQA), and LongBench, consistently showing improvements over RoPE and NoPE baselines.
- **Weaknesses:**
  - While the paper compares REPO against RoPE and NoPE, further comparisons with other dynamic position methods like COPE (beyond just a textual discussion) would have strengthened the empirical claims regarding efficiency and performance.
  - While the authors claim $f_{\phi}$ learns positions in a "denser and non-linear" space, the exact mechanism of how the model "interprets" these real-valued shifts in the RoPE rotation framework could be explored more deeply.

#### **Presentation**

- **Strengths:**
  - The paper is exceptionally well-written. The transition from the cognitive science motivation to the mathematical implementation (Equations 4–7) is logical and easy to follow.
  - The use of statistics for positional distances and attention mass distribution (Figure 2 and Table 4) effectively supports the authors' claims about how REPO breaks locality bias.
- **Weaknesses:**
  - While the layer threshold (1/3-th layer) is mentioned, more intuition or an ablation in the main text (rather than just the Appendix) on why lower layers are excluded would benefit the narrative flow.

#### **Significance**

- **Strengths:**
  - Managing long context and structured data is a central challenge in LLM research. REPO appears potentially complementary to downstream paradigms such as RAG and agentic systems.
  - The module is lightweight, introducing negligible overhead ($3.84 \times 10^{20}$ vs $3.87 \times 10^{20}$ FLOPS), making it a practical candidate for production-scale models.
- **Weaknesses:**
  - The requirement for continual pre-training might limit its adoption for users who rely solely on prompt engineering or fine-tuning on limited data.

#### **Originality**

- **Strengths:**
  - Predicting positions directly from hidden states $h_i$ via a SwiGLU sub-layer and integrating them into a differentiable rotation matrix is a creative and efficient way to achieve dynamic context management.
  - Moving beyond "counting" or linear ordering to a content-aware position assignment is a significant conceptual shift from standard Transformer architectures.

---

> ### Author Rebuttal · Authors · 2026-03-30
>
> > **Q1: ... further comparisons with other dynamic position methods like COPE (beyond just a textual discussion) would ...**
>
> Thank you for your suggestion. We actually have spent a lot of effort on the COPE baseline. Firstly, the CoPE method requires the exact attention score for computing the position value, which is unsupported by Flash Attention. Training with a naive attention implementation is a disaster for LLMs with more than 1B parameters, leading easily to OOM. To fix the OOM issue, we set the batch size to 1 and enable activation checkpointing to save memory usage on H100s. It takes around 2 weeks to fine-tune the 1B model with CoPE on midsize training data. Secondly, unlike RePo which works compatibly with almost all the position encoding methods, CoPE is actually another position encoding method. We train CoPE under a setup similar to other baselines to ensure a fair comparison. However, directly fine-tuning CoPE on the pre-trained 1B model causes more severe loss spikes, and the final performance of CoPE is worse than all the baselines. We adopt the official code from the CoPE paper for the implementation. We will release the code for CoPE and its configuration together with our codebase so that others can reproduce our results.
>
>
> > **Q2: While the authors claim f_phi learns positions in a "denser and non-linear" space ... could be explored more deeply.**
>
> We appreciate your valuable suggestion for more explanation about the learned positions. Under the rotation framework of RoPE, the fixed training context length only covers a small portion of the period of low-frequency dimensions. Thus, the extrapolation for those dimensions would be more difficult, i.e., a more severe out-of-distribution issue. However, the learned denser and non-linear space can better leverage the small portion of the training context length to achieve better long-context performance. Thank you again for your suggestion and we will integrate this discussion into our revised paper.
>
>
> > **Q3: While the layer threshold (1/3-th layer) is mentioned, more intuition or an ablation in the main text (rather than just the Appendix) on why lower layers are excluded would benefit the narrative flow.**
>
> Thank you for the feedback. The discussion for the intuition of the layer threshold is discussed in Section 4.2 (Training & Efficiency) between lines 149 and 180. In other words, this design choice is motivated by previous findings that lower layers of LLMs primarily capture surface-level features that depend more on local information. We will highlight this intuition more clearly in the revised paper.
>
> > **Q4: The requirement for continual pre-training might limit its adoption for users who rely solely on prompt engineering or fine-tuning on limited data.**
>
> Thanks for your concern. We choose to continually pretrain exsiting LLM checkpoints for two reasons: (1) we cannot train a model from scratch due to limited compute resources, and (2) more importantly, we want to show that any exsiting LLM checkpoint can be enhanced by the our method quickly. Note that the continual pretraining uses only general-purpose data. Once the training is complete, the resulting checkpoints can be used for any downstream tasks, just like the original checkpoints. Therefore, our method introduces no additional difficulty when adapting to different application scenarios, such as prompt engineering or fine-tuning on limited data.
>
> > **Q5: The experiments are conducted on 1B and 7B models. Do the authors expect the benefits of REPO to scale similarly to larger models (e.g., >30B), where positional inductive biases may interact differently with attention patterns?**
>
> A big advantage of RePo is that all the patterns are unsupervisedly learned from the model’s perspective, rather than pre-defined from huamn experts. Thus, when scaling up the model and data size, the model has high potential to find a more appropriate re-positioning function that fits for real-world applications. It is worth noting that our RePo method has the potentaion to degrade to the standard linear position assignment (Appendix A), we believe RePo can learn a function that no worse than the standard one. In addition, experiments show that the performance gain on structured data and long-context tasks remains to be significant when scaling from 1B to 7B models. So we expect similar benefits to generalize to larger model and data scales.
>
> > **Q6: How sensitive is the performance to the  1/8 hidden size ... or is position truly a low-dimensional signal?**
>
> Thanks for your very interesting question. Due to the limitation of computational resources, in our preliminary study, we only conducted experiments on $d_p \in \{1, d/8\}$.  Though setting  $d_p = d/8$ achieves better performance, setting  $d_p = 1$ already shows very interesting patterns when analyzing the model. We will conduct more analysis in our future work to analyze the effect of position representation.

---

> > ### Author Rebuttal · Reviewer_uefV · 2026-04-01
> >
> > My concerns have been addressed, so I will keep my score as it is.

---

### Official Review · Reviewer_RMYH · 2026-03-13

**Soundness:** 2
**Presentation:** 2
**Significance:** 2
**Originality:** 3
**Overall Recommendation:** 4
**Confidence:** 4

**Summary:**

This paper proposes REPO, a lightweight module that replaces fixed linear token positions with learned continuous positions based on hidden states. In the paper the method is applied during continual pretraining on OLMo-2 1B and 7B, and the paper reports improvements on noisy-context, structured-data, and long-context benchmarks.

**Compliance With Llm Reviewing Policy:**

Affirmed.

**Final Justification:**

Most of my concerns were addressed. I will keep my score.

**Key Questions For Authors:**

1. Can the authors show that the positions learned by RePo are different from the effective positional behavior of RoPE / RoPe with vanilla learned positional embeddings, both qualitatively and quantitatively?
2. Please improve Figure 3 by including concrete visual examples of “constant”, “mono” and “hybrid” patterns, and clarify why this taxonomy is meaningful.
3. Can the authors provide more compelling motivation for the CLT-based claims?

**Limitations:**

Yes

**Strengths And Weaknesses:**

**Strengths:**

1. The paper studies an important problem: whether fixed positional structure limits LLM performance, especially for noisy or long contexts.
2. The proposed method is conceptually interesting and appears simple to integrate with existing positional encoding methods such as RoPe.
3. The paper includes experiments on several relevant benchmarks (RULER, NIAH, HybridQA, LongBench), and some results suggest potential benefits on context-heavy tasks.
4. The authors analyze learned position behavior.

**Weaknesses:**

1. The connection between Cognitive Load Theory (CLT) / human working-memory is weakly tied to the technical contribution. It would be better if the authors motivate the positional encoding problem directly in terms of LLM inductive bias, attention behavior, and length generalization.
2. A key missing validation is whether the learned position embeddings are meaningfully different from standard RoPe behavior / RoPe with vanilla learned positional embeddings. It would be best if you can show how the learned RePo positions are different in practice.
3. Figure 3 is hard to interpret, and the paper does not show concrete visual examples of what “constant,” “mono,” and “hybrid” patterns look like.
4. The claim in Section 5.2, “the assigned positions can capture the intrinsic structure of the input context”, is somewhat vague, and the connection to CLT appears to be weak.

**Minor:**

1. Table 5 should bold the best method for each column, consistent with other tables.
2. Table 4 should clearly explain what the 10^(-2) scaling means.

---

> ### Author Rebuttal · Authors · 2026-03-30
>
> > **Q1: The connection between Cognitive Load Theory (CLT) / human working-memory is weakly tied to the technical contribution. It would be better if the authors motivate the positional encoding problem directly in terms of LLM inductive bias, attention behavior, and length generalization.**
>
> Thanks for your helpful feedback. Our analysis experiments, e.g., the improved attention allocation by RePo in Sec 5.1, are designed to connect with CLT. We will strengthen this connection more clearly and highlight the technical problems you mentioned in the revised paper.
>
> > **Q2: A key missing validation is whether the learned position embeddings are meaningfully different from standard RoPe behavior / RoPe with vanilla learned positional embeddings. It would be best if you can show how the learned RePo positions are different in practice.**
>
> Thank you for the constructive suggestion. Due to the limited page length, our investigation on the unique behaviors of RePo was presented in Appendix C & D. In contrast to the static position in RoPE, we have two main observations: (1) RePo assigns patterns that capture the intrinsic structure of data;  (2) The position patterns are different for different attention heads and data input.
>
> In Appendix C, we evaluate the behaviors of RePo on a synthetic task, namely text reversal. Given input “A B C”, the LM needs to generate “C B A”. This task is selected on purpose, because the distance between a generated token and its corresponding dependent input token grows linearly as the generation proceeds. We find that the RePo method learns to assign closer positions for pairs of tokens.
>
> In Appendix D, we conducted a case study on the positions learned by RePo. Interestingly, we find that the RePo can adaptively assign positions roughly aligning with the semantic segmentations of the input context, e.g., the prompt and question-answer pairs in Figure 8. We observe that this emerging ability is not limited to specific data, and displays interesting patterns in other data structures, e.g., tables and graphs. Since those structures are learned by the model in an unsupervised way, we believe they are more beneficial to the prediction of the model.
>
> In both sections, we find that RePo learns head-specific position patterns. When the model is trained on general data, i.e., the mid-training data used for OLMo-2, RePo also learns different position patterns for different input data.
>
> We promise to highlight this part in the main paper if we have more space.
>
> > **Q3: Figure 3 is hard to interpret, and the paper does not show concrete visual examples of what “constant,” “mono,” and “hybrid” patterns look like.**
>
> Thanks for your question. We agree that concrete examples would better illustrate the pattern of “constant”, “mono”, and “hybrid”, and ease the understanding on this part. As shown in Figure 8 (Appendix D), the three sub-figures are actually selected on purpose to demonstrate positions assigned by RePo that align with the three types of patterns. As shown in Figure 8 (a), it demonstrates the constant pattern, where positions are assigned in a fixed range $ [a-\epsilon, a+\epsilon]$, which mimics the NoPE method as discussed in Appendix A. In contrast, as shown in Figure 8 (c), the positions assigned by RePo increase almost monotonically, which is similar to the standard RoPE method. In Figure 8 (b), there exist many parts that have non-linear positions, i.e., the hybrid patterns. We will follow your suggestion to strengthen this connection in the revised paper.
>
>
> > **Q4: The claim in Section 5.2, “the assigned positions can capture the intrinsic structure of the input context”, is somewhat vague, and the connection to CLT appears to be weak.**
>
> Thanks for your valuable suggestion. As we mentioned in Q2, please refer to the Case Study section (Appendix D) for more detailed explanation.  We find that RePo can adaptively assign positions roughly aligning with the semantic segmentations of the input context, e.g., the prompt and question-answer pairs. We observe that this emerging ability is not limited to specific data, displaying interesting patterns in other data structures, e.g., table and graph.
>
> > **Minor**
>
> Thanks for pointing out those issues, and we will follow your suggestion to fix all of them.

---

> > ### Author Rebuttal · Reviewer_RMYH · 2026-04-03
> >
> > Thank you for your rebuttal.
> > Most of my concerns were addressed. I will keep my score.

---

### Decision · Program_Chairs · 2026-04-30

**Decision:**

Accept (regular)

**Comment:**

In this paper the authors introduce a context re-position approach (RePo) to deal with the limitations of fixed linear positional encoding commonly used in LLMs. Inspired by the Cognitive Load Theory, RePo is a lightweight network that learns continuous positions based on hidden states.  Extensive experiments are carried out using OLMo-2 1B & 7B models on a variety of benchmarks. The results demonstrate superior performance of RePo over some existing positional encoding techniques. All reviewers find the idea interesting with sufficient novelty. Most of the concerns on motivation (e.g. how to position the Cognitive Load Theory with respect to this work) and some technical details have been successfully addressed in the rebuttal. Overall, this is an interesting work and may have its value to the machine learning community. The authors need to revise the paper accordingly in its final version.